# Impacts of Climate Warming and Humidification on Vegetation Activity over the Tibetan Plateau



Zhe He [1,†], Ting Zhou [2,†], Jiaqi Chen [1], Yajing Fu [1], Yuanying Peng [3], Li Zhang [1,4], Tongyu Yao [1], Taimoor Hassan Farooq [5], Xiaohong Wu [1,4], Wende Yan [1] and Jun Wang [1,*]

1   Technology in Forestry and Ecology in South China, Central South University of Forestry and Technology, Changsha 410004, China; hezhe011006@gmail.com (Z.H.); jc373069@gmail.com (J.C.); 2419131345f@gmail.com (Y.F.); woshizl1989@126.com (L.Z.); genewar661@gmail.com (T.Y.); wxh16403@163.com (X.W.); t20001421@csuft.edu.cn (W.Y.)
2   China International Engineering Consulting Corporation, Ecological Technical Research Institute, Beijing 100085, China; zt@ciecc.com.cn
3   College of Arts and Sciences, Lewis University, Romeoville, IL 60446, USA; pengyu@lewisu.edu
4   Lutou National Station for Scientific Observation and Research of Forest Ecosystem in Hunan Province, Yueyang 414000, China
5   Bangor College China, A Joint School between Bangor University and Central South University of Forestry and Technology, Changsha 410004, China; t.farooq@bangor.ac.uk
*   Correspondence: jwang0829@csuft.edu.cn; Tel./Fax: +86-731-85623868
†   These authors contributed equally to this work.

**Abstract:** Vegetation is the most vulnerable component of terrestrial ecosystems to climate change. In recent decades, there has been a significant warming and humidification trend in the Tibetan Plateau. It is crucial to study and analyze the impact of these changes on the ecosystem and their future trends for protecting the Tibetan Plateau's ecosystem. This study collected and analyzed climate (temperature, precipitation) data and vegetation index data (the normalized difference vegetation index (NDVI) and the leaf area index (LAI)), as well as data from significance tests combined with Mann–Kendall tests and Sen's slope estimation. The effects of temperature and precipitation factors on vegetation indices were revealed, leading to a multiple regression model predicting NDVI and LAI value changes under climate change from 2021 to 2100. The results indicate a general increase in temperature and precipitation levels across the Tibetan Plateau between 2000 and 2020. The climate experienced a clear pattern of warming and moist conditions, with the southeast region experiencing warmer and wetter conditions, and the northwest region experiencing drier and colder conditions. The trends of the LAI and NDVI values of the Tibetan Plateau indicated a general increase, with a gradual decline from the southeast to the northwest. Precipitation and temperature were differentially correlated with the NDVI and LAI values across various regions of the plateau. Between 2021 and 2100, the Tibetan Plateau is expected to experience year-on-year increases in both precipitation and temperature levels. However, the increase in precipitation was found to be less significant than that of the climate and, comparatively, smoother. There is a certain correlation between the NDVI and LAI values, and the changes in temperature and precipitation. The variations of both are more influenced by temperature than precipitation, with an overall increasing trend observed over the years, which is also quite evident. This study could serve as a scientific foundation and a point of reference for monitoring vegetation changes over a long period of time on the plateau, as well as for the planning and execution of ecological development in the Tibetan Plateau.

**Keywords:** warming and humidification; NDVI; LAI; Tibetan Plateau

## 1. Introduction

In recent years, we have observed major changes in the global climate, in the active voice. According to [1], the global climate system is projected to continue warming without

a significant slowdown until the middle of the century. High-altitude or high-latitude regions are particularly affected by global warming [2], including the Tibetan Plateau, which is the highest plateau in the world. The Tibetan Plateau has experienced significantly more pronounced global warming than other regions, with temperatures rising about 30 years ahead of the global warming curve and at twice the rate of average land-based global warming. [3] This worrying trend is damaging the Tibetan Plateau's glaciers and snow cover, causing them to gradually shrink and ultimately increasing the water cycle. Since the 1880s, rainfall on the Tibetan Plateau has displayed a fluctuating trend of overall increase [4]. Compared with other regions, the Qinghai–Tibet Plateau is more affected by climate warming and humidification, with a significant increase in the frequency of extreme-high-temperature events and a significant decrease in the frequency of extreme-low-temperature events [5,6]. Therefore, the scientific and effective evaluation of the impacts of climate warming and humidification on the ecological environment of the Tibetan Plateau, as well as the exploration of change patterns, have become the focus of attention of many researchers [7]. Due to its unique and fragile ecological environment and climate sensitivity, the Tibetan Plateau is regarded as a "climate change sensor" for Asia, and even the entire hemisphere [8].

Global warming has had a profound impact on the terrestrial ecosystems of the Tibetan Plateau at multiple scales and over long periods of time, including a reduction in wetland and grassland areas, serious land desertification, increased natural disasters, and the endangerment of many flora and fauna species, which have become more and more prominent [9]. In terrestrial ecosystems, vegetation, as a natural link between soil, the atmosphere, and water, is sensitive to ecological environmental changes [10]. On the one hand, changes in the water and heat environment caused by climate change will have an impact on the climatic characteristics of vegetation, and change the regional distribution characteristics of vegetation as well as the functions and processes of vegetation ecosystems. On the other hand, through physiological activities such as photosynthesis, respiration, and transpiration, vegetation regulates water balance, carbon balance, energy exchange, and the stability of the regional climate of land ecosystems [11,12], which are the basis and key to the balance and stability of terrestrial ecosystems. Thus, studying vegetation change is crucial for understanding the ecological and environmental impacts of climate warming on the Tibetan Plateau, which is essential for maintaining ecosystem balance and stability.

Changes in vegetation growth, development, and distribution can be a reflection of the effects of climate change. The commonly used vegetation data at this stage can be broadly divided into those of the NDVI and those of the LAI. The NDVI is an essential indicator of vegetation cover and growth status when using remote sensing imagery, making it an effective monitoring tool for assessing vegetation's response to climate change [13]. In addition to reflecting the growth and distribution of vegetation on a global or regional scale, it could also characterize trends of change over long periods of time [11]. When analyzed in combination with the LAI, a comprehensive indicator of vegetation's use of light energy and canopy structure [14], the NDVI can directly reflect the growth status of vegetation. In recent years, with the development of remote sensing technology and the improvement of radiative transfer mode, LAI parameter estimation has become more accurate. The leaf orientation, leaf-to-wood ratio, and optical properties of different vegetation will lead to differences in leaf density, leaf inclination, leaf area scattering, leaf reflection coefficients, transmittance coefficients, etc., which will further affect LAI values; therefore, the type of vegetation determines the characteristics of the LAI to a certain extent [15]. In conclusion, analyzing the temporal and spatial patterns of the NDVI and LAI values of vegetation, and their correlations with climate variables, can provide insights into the response mechanisms of vegetation to climate change.

In the context of climate warming and humidification, studying the changes in vegetation and the factors influencing the climate on the Tibetan Plateau is essential [16]. The main climate factors driving vegetation change on the Tibetan Plateau are temperature and precipitation, which influence vegetation growth and distribution through their effects on

respiration, photosynthesis, and soil organic carbon depletion. In recent years, numerous researchers across the globe have conducted extensive studies on the dynamic fluctuations of vegetation at different times and spaces, and the relationship between vegetation and temperature, precipitation, and other climate factors using the NDVI. Some researchers believe that vegetation changes on the Tibetan Plateau are mainly affected by temperature; e.g., [17], Zhang Goli et al. analyzed GIMMS-NDVI data and concluded that vegetation on the Tibetan Plateau is more sensitive to temperature changes than to precipitation [18]. Other researchers believe that vegetation changes on the Tibetan Plateau are mainly affected by precipitation; e.g., Huang et al. analyzed a 17-year NDVI trend on the Tibetan Plateau using MOD09A1, data and discovered that, while the delay of the NDVI values on temperature is not as evident, the delays on precipitation, relative humidity, and radiation are more evident [19]. Other researchers believe that vegetation changes on the Tibetan Plateau are the result of the combined effects of temperature and precipitation; e.g., Huang et al. demonstrated that increases in temperature and precipitation favor the growth of vegetation, but the combined effect of the two exhibits strong spatial heterogeneity [20]. Both temperature and precipitation have different degrees of influence on vegetation change, and the response of vegetation to climate change is complex, mainly due to the scale of the Tibetan Plateau and the type and distribution of vegetation. Therefore, conducting an extensive investigation into vegetation and its reaction to climate warming and humidification within a vast area could aid us in gaining a profound comprehension of the spatiotemporal patterns of climate alterations on the vegetation of the Tibetan Plateau.

In summary, the ecological environment of the Tibetan Plateau is very fragile, and the vegetation type and distribution are significantly affected by climate warming and humidification. Currently, there are few studies on the relationship between vegetation and climate warming and humidification on the Tibetan Plateau. Therefore, firstly, this paper analyzes the impact on ecosystems in the Tibetan Plateau under the trend of climate warming and humidification as well as their future trends. Secondly, this paper reveals the spatial and temporal characteristics of vegetation in the Tibetan Plateau region and their relationship with climatic factors, and provides references to the long time series study of changes in the Tibetan Plateau's vegetation. Finally, we summarize the spatial and temporal variation patterns of vegetation and climate events in the Tibetan Plateau region through data analysis, and predict trend changes in the ecological environment of the plateau. This is conducive to improving monitoring and earlywarning capabilities with regard to related climate disasters, and providing a scientific basis for the rational planning of the sustainable development and utilization of the Tibetan Plateau, as well as the effective management of its ecosystems.

## 2. Materials and Methods

### 2.1. Study Site

The Tibetan Plateau (73°19′ E–104°45′ E; 26°00′ N–39°47′ N) is located in southwestern China and mainly includes the Tibet Autonomous Region, Qinghai Province, and parts of the four provinces of Sichuan, Yunnan, Gansu, and Xinjiang Uygur Autonomous Region, as shown in Figure 1. The Tibetan Plateau encompasses a vast expanse and boasts a variety of climate types. The average altitude of the region is above 4000 m [21], and its total area is about $257.24 \times 10^4$ km$^2$. Due to its unique climate type, the Tibetan Plateau is defined separately as the "Tibetan Plateau Climate Region" in the whole climate zone. The climate is cold and dry throughout the year, and the spatial distributions of precipitation and temperature are significantly different. From southeast to northwest, the average annual temperature ranges from −5 to 15.5 °C, and the annual cumulative precipitation ranges from 16 to 1764 mm [22,23] Under the joint nurturing of topography and climate, the Tibetan Plateau has formed alpine vegetation types, including deserts, grasslands, scrubs, and meadows [24]. The ecological environment of the Tibetan Plateau is highly fragile and sensitive due to the unique climatic characteristics of the region, and its ecological recovery and self-regulation capacities are poor [25].

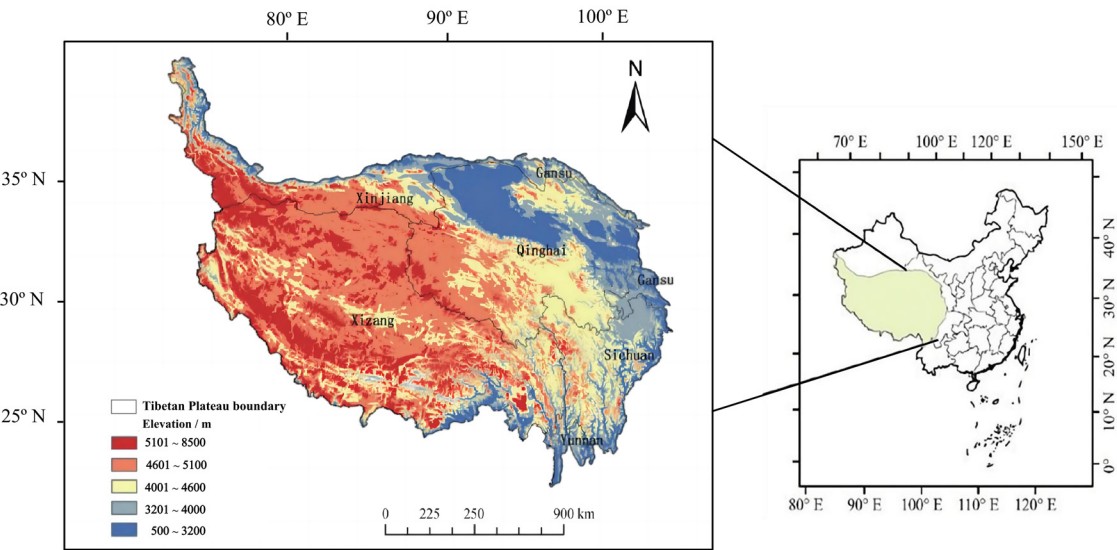

**Figure 1.** The location and elevation of the Tibetan Plateau.

*2.2. Date Sources and Preprocessing*

2.2.1. NDVI Data

We obtained NDVI data from a spatially resolved 250 m, 16-day synthetic L3 data product (MOD13Q1) provided by NASA's data center (https://ladsweb.modaps.eosdis.nasa.gov/ (accessed on 22 September 2022)) [26]. The MOD13Q1 product of the Tibetan Plateau region from 2000 to 2020 was batch-downloaded using the Internet Download Manager (IDM) tool, and the NDVI sub-dataset of the MOD13Q1 data was extracted using the PyCharm tool (a tool for integrated development environments, designed for the Python language). The NDVI data were converted from an HDF format to a TIF format, after which block mosaics and boundary cropping were performed in ArcGIS to obtain the Tibetan Plateau's regions. The raster data were then preprocessed using the model builder method (an application for creating, editing, and managing models) in ArcGIS, including batch projection transformation, cropping, and resampling. Finally, the data were maximally synthesized into annual data using the maximum value composite (MVC, a method for analyzing the trend and causes of vegetation coverage change using the annual maximum NDVI value).

2.2.2. LAI Data

We obtained LAI data from the L4 data product (MOD15A2H) [27], which has a spatial resolution of 500 m and an 8-day synthesis. The data were acquired from the National Aeronautics and Space Administration (NASA) data center (https://ladsweb.modaps.eosdis.nasa.gov/ (accessed on 22 September 2022)). The processing of the LAI data followed a similar procedure to that used for the previously described NDVI data.

2.2.3. Historical Meteorological Data

Historical meteorological data were obtained from the 1901–2022 China's 1 km resolution month-by-month mean air temperature dataset [28] and the 1901–2022 China's 1 km resolution month-by-month precipitation dataset [29], courtesy of the National Tibetan Plateau Data Center (TPDC) (https://data.tpdc.ac.cn/ (accessed on 9 October 2022)). After batch-downloading the dataset, PyCharm was used to convert the temperature and precipitation data from the NC format to the TIF format, and then a series of operations such as cropping and reprojection were carried out in ArcGIS. Using the image statistical tool of ArcGIS, the monthly temperature and precipitation data were calculated and processed into annual mean temperature data and annual cumulative precipitation data, respectively.

### 2.2.4. Future Meteorological Data

The future meteorological data were obtained from China's 1 km resolution multi-scenario multi-model month-by-month mean temperature dataset from 2021 to 2100 [30] and China's 1 km resolution multi-scenario multi-model month-by-month precipitation dataset from 2021 to 2100, both courtesy of the TPDC (https://data.tpdc.ac.cn/ (accessed on 9 October 2022)) [31]. The future meteorological data were processed in the same way as the historical meteorological data mentioned above. The data were produced using the delta spatial downscaling program, which is an algorithm that maps data from high resolution to low resolution, in China [32]. The data were based on the global climate model dataset, released by the IPCC Coupled Model Intercomparison Program Phase 6 (CMIP6), which has a resolution greater than 100 km, as well as the global high-resolution climate dataset released by WorldClim 1.4.

### 2.3. Methods

#### 2.3.1. Trend Significance Test Analysis

In this study, we used a trend significance test by combining the Mann–Kendall (MK) test and the Theil–Sen median method (Sen's slope estimation) in order to examine the trend and significance of climate alteration on the Tibetan Plateau from 2000 to 2020 in combination. The study revealed the impacts of precipitation and temperature factors on the vegetation index of the ecosystem. Simultaneously, the trend prediction of climate development and its impact on the Tibetan Plateau for the period from 2021 to 2100 were analyzed. The MK test is a nonparametric test [33] that is employed to evaluate the trend of climatic factors in a time series, eliminating the need for measurements to follow a normal distribution. Additionally, the test is not affected by missing values and outliers, and has been widely used in climate analysis [34]. Sen's slope estimation method is a reliable technique for calculating climatic variable gradients, and is suitable for analyzing trends in long time series of data [35]. The statistical significance of the NDVI values, LAI values, and climatic trends was assessed using the MK test, while Theil–Sen median was used to determine the magnitude of trends in these variables. The Mann–Kendall test was performed as follows:

$$Z = \begin{cases} \frac{S-1}{\sqrt{VAR\ (S)}}, & S > 0 \\ 0, & S = 0 \\ \frac{S+1}{\sqrt{VAR(S)}}, & S < 0 \end{cases}$$

$$VAR(S) = \left\{ n(n-1)(2n+5) - \sum_{j}^{p} t_j(t_j - 1)(2t_j + 5) \right\} \div 18$$

where $S$ is the time series data, $p$ is the number of nodes in the dataset, and $t_j$ is the length of the node. We set the significance level $\alpha = 0.05$ and accepted the original hypothesis when $|Z| \leq Z1 - \alpha/2$. Otherwise, the hypothesis was rejected, indicating a significant trend change.

Sen's slope was estimated as follows:

$$\beta = median\left(\frac{x_j - x_i}{j - i}\right) \forall 1 < i < j < n$$

where $x_j$ and $x_i$ are the time series data, $\beta$ is the rate of change in the slope, $\beta > 0$ indicates that the time series shows an upward trend, and $\beta < 0$ indicates that the time series shows a downward trend.

The Sen + MK trend analysis methods, using the R language for raster calculation and Sen + MK calculation, and the trend change division based on the calculation results, were as follows:

When $S > 0$, $|Z| < 1.96$, there was a significant upward trend observed. However, when $S > 0$, $|Z| > 1.96$, the upward trend was considered non-significant. Conversely,

when $S < 0$, $|Z| < 1.96$, the downward trend was non-significant. However, when $S < 0$, $|Z| > 1.96$, a significant downward trend was observed.

### 2.3.2. Correlation Analysis

A Pearson's correlation analysis of the NDVI and LAI's climate drivers was performed using the SPSS 13.0 software [36], and a *t*-test was used to test the significance of the correlation coefficients [37] to better understand the response relationship between the ecosystems of the Qinghai–Tibet Plateau, the climate drivers, and the spatial distribution correlation.

### 2.3.3. Multiple Linear Regression Model

A multiple linear regression model (MLRM) was used to predict the trend of ecosystem changes on the Tibetan Plateau due to climate change from 2021 to 2100. An important assumption of the multiple linear regression model was that the error obeys the normal distribution [38], and the frequency histograms of the NDVI and LAI values were analyzed and found to be bell-shaped, which indicated that the NDVI and LAI data were normally distributed. On the basis of analyzing the correlation between precipitation, temperature, and the NDVI and LAI values in the historical data, a linear regression model was developed between them, which was used to predict the trends of the NDVI and LAI values in the predicted period (2021–2100), based on the changing trends of the climatic factors.

## 3. Results

### *3.1. Climate Change on the Tibetan Plateau from 2000 to 2020*

#### 3.1.1. Variation Trend of Precipitation of the Qinghai–Tibet Plateau from 2000 to 2020

From 2000 to 2020, the Tibetan Plateau experienced a downward trend in maximum precipitation, with the minimum of 2626.2 mm occurring in 2009 and the peak of 3677.0 mm recorded in 2010 (Figure 2). The annual average precipitation ranged from 3.7 mm to 3141.6 mm, showing a significant spatial distribution difference. Most regions experienced low annual precipitation, with only a few areas in the south receiving substantial annual rainfall (Figure 3a). Based on Figure 3b, the annual precipitation in the Tibetan Plateau over the past 20 years has generally increased in most regions, but decreased in a few southern areas. Over the past two decades, significant precipitation trends have been observed in most eastern Tibetan Plateau areas, while changes in most western regions have been negligible. Specifically, there has been a noteworthy increase in precipitation in small areas in the eastern part of the plateau, while some regions in the southern part have experienced a significant decrease (see Figure 3c,d).

#### 3.1.2. Trend of Temperature Change in the Tibetan Plateau from 2000 to 2020

From 2000 to 2020, there was a general upward and warming trend of temperature change on the Tibetan Plateau. The lowest recorded temperature of −2.5 °C was in 2000, while the highest recorded temperature of −1.4 °C was in 2009, as evidenced in Figure 4. The multi-year average temperature range on the Tibetan Plateau from 2000 to 2020 was −31.4 °C to 24.7 °C, with significant heterogeneity across the region. The Tibetan Plateau generally experiences low temperatures in most regions, with higher temperatures found in the southeast and some northern areas. The temperature gradually decreases from the southeast to the northwest (Figure 5a). The central and northeastern parts of the plateau exhibited a notable increasing temperature trend, which was more pronounced, as shown in Figure 5b,d. Overall, the climate change in the Tibetan Plateau from 2000 to 2020 was significant (Figure 5c).

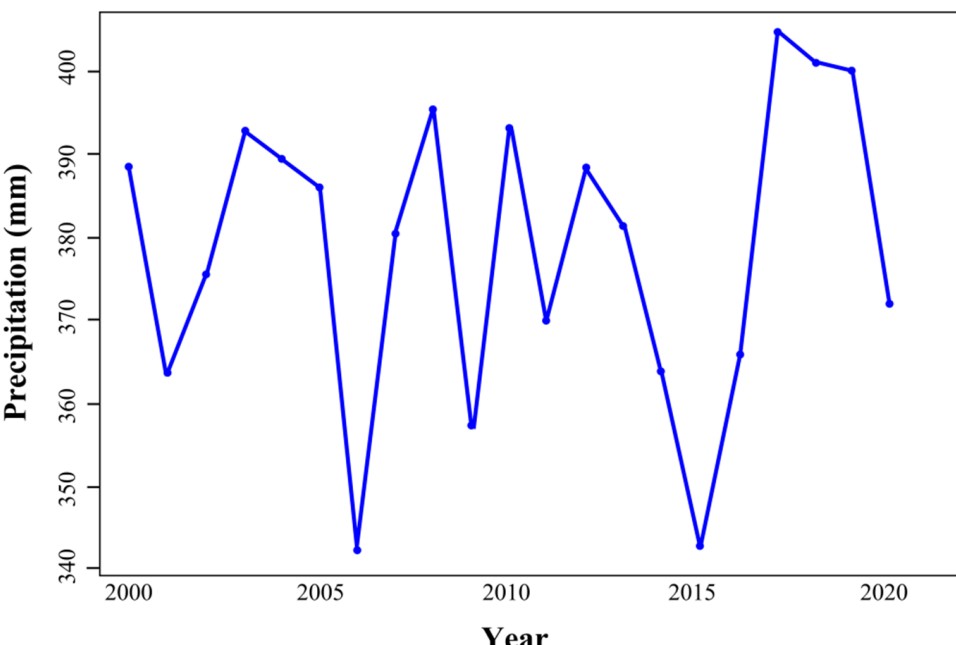

**Figure 2.** Changes in precipitation over the Tibetan Plateau during 2000–2020.

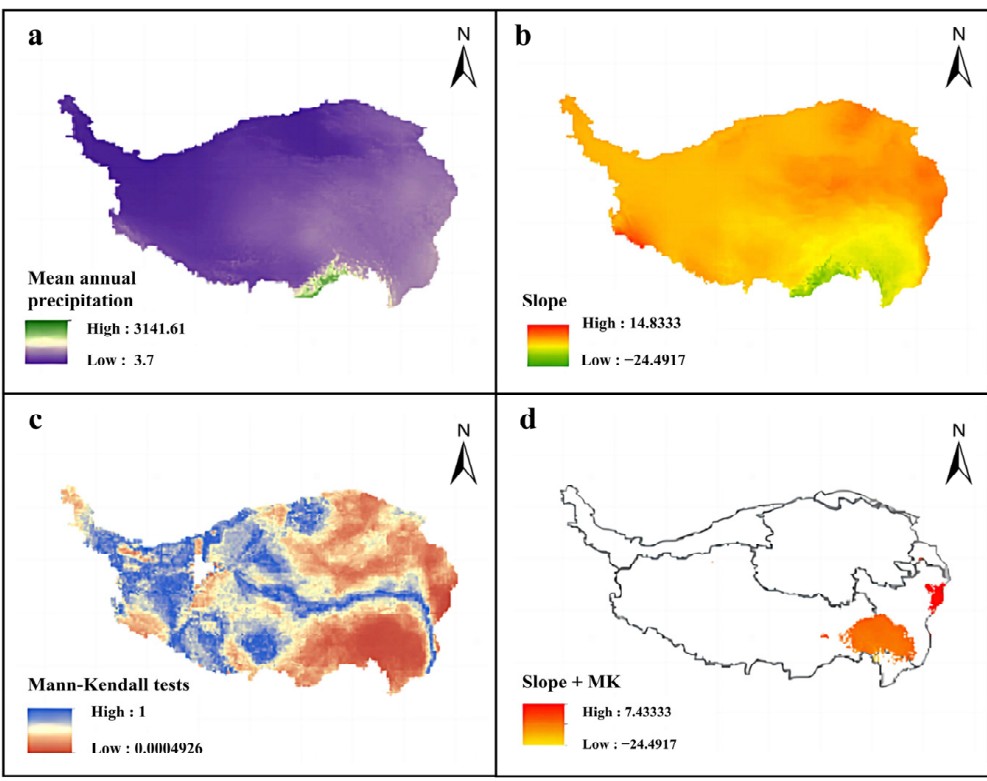

**Figure 3.** Spatiotemporal variation of precipitation over the Tibetan Plateau during 2000–2020. (**a**) Spatial distribution of annual average rainfall over the Tibetan Plateau during 2000–2020. (**b**) Sen's slope estimation of rainfall over the Tibetan Plateau during 2000–2020. (**c**) Mann–Kendall tests of rainfall over the Tibetan Plateau during 2000–2020. (**d**) The significance of rainfall changes over the Tibetan Plateau during 2000–2020.

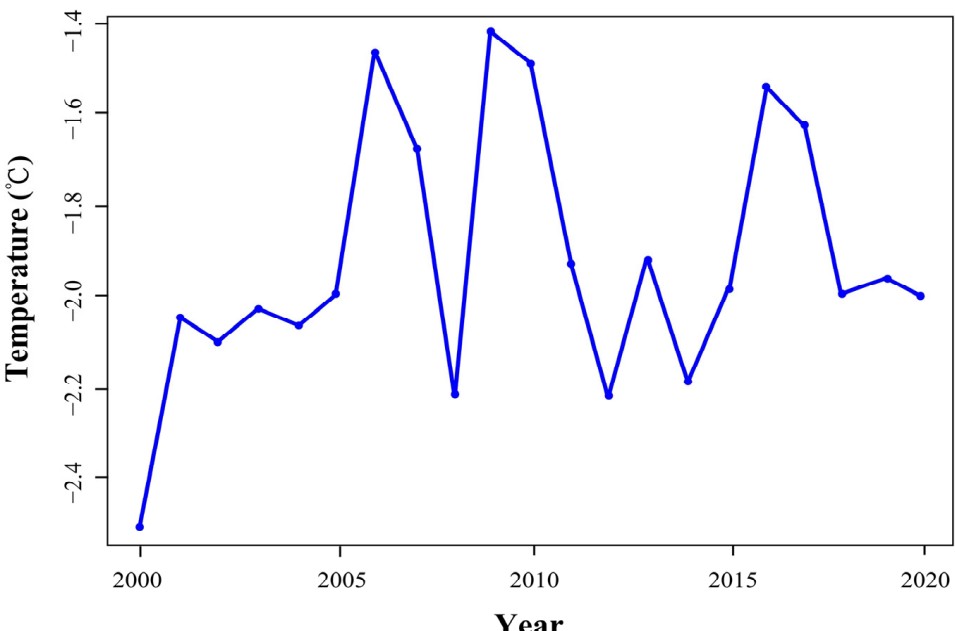

**Figure 4.** Changes in temperature over the Tibetan Plateau during 2000–2020.

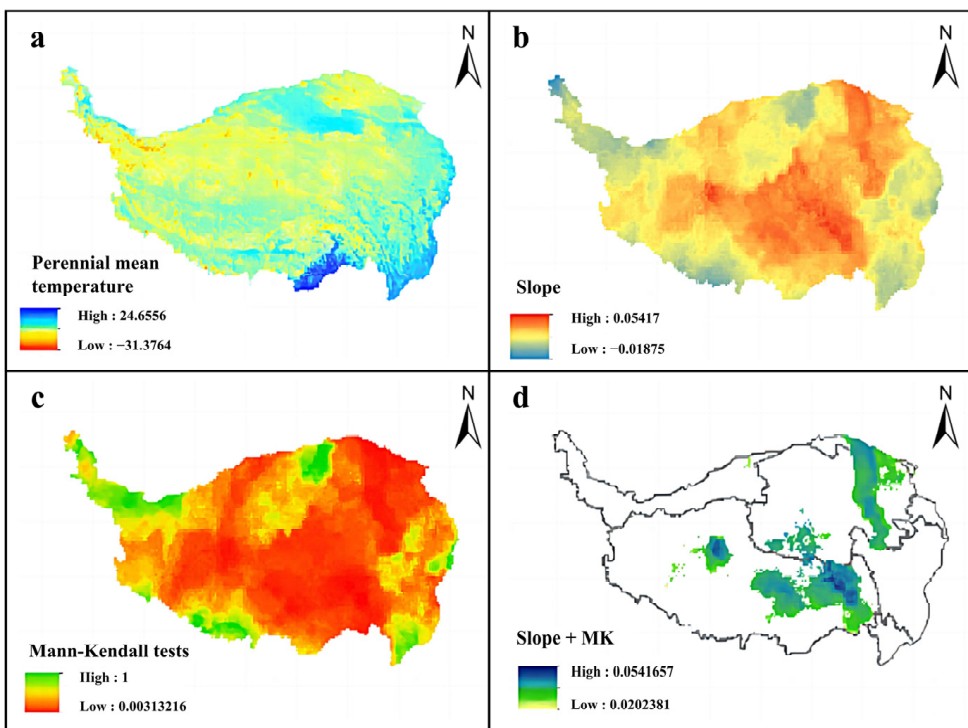

**Figure 5.** Spatiotemporal variation of temperature over the Tibetan Plateau during 2000–2020. (**a**) Spatial distribution of mean temperature over the Tibetan Plateau during 2000–2020. (**b**) Sen's slope estimation of temperature over the Tibetan Plateau during 2000–2020. (**c**) Mann–Kendall tests of temperature over the Tibetan Plateau during 2000–2020. (**d**) The significance of temperature changes over the Tibetan Plateau during 2000–2020.

*3.2. Changes in the Tibetan Plateau's Ecosystem from 2000 to 2020*

3.2.1. Trend of NDVI Change in the Qinghai–Tibet Plateau from 2000 to 2020

During the period from 2000 to 2020, the changes in the NDVI values in the Tibetan Plateau generally showed an increasing trend, and during the study period, the NDVI values of 2005 and 2008 were the lowest at 0.1798 and 0.1790, respectively. Subsequently,

there was a significant increase in the NDVI value in 2008 (Figure 6). For many years, the spatial heterogeneity of the multi-annual mean NDVI values over the Tibetan Plateau was obvious, with the mean NDVI values ranging from −0.1484 to 0.8547, and the NDVI showed a gradual decline from the southeast to the northwest on the whole (Figure 7a). In general, the NDVI essentially showed an upward trend, and the overall change was significant, with a significant upward trend in most regions (Figure 7b–d).

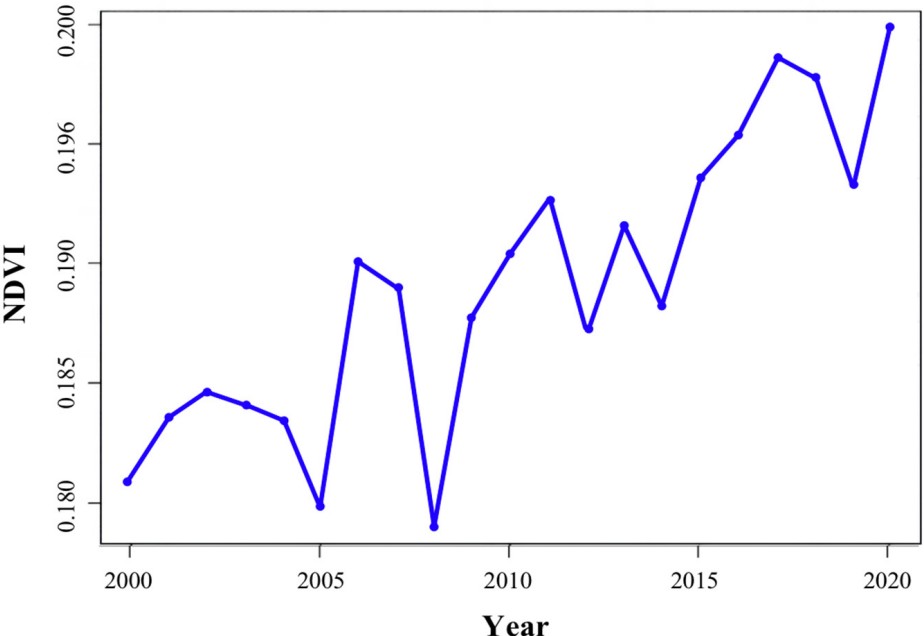

**Figure 6.** Changes in the NDVI over the Tibetan Plateau during 2000–2020.

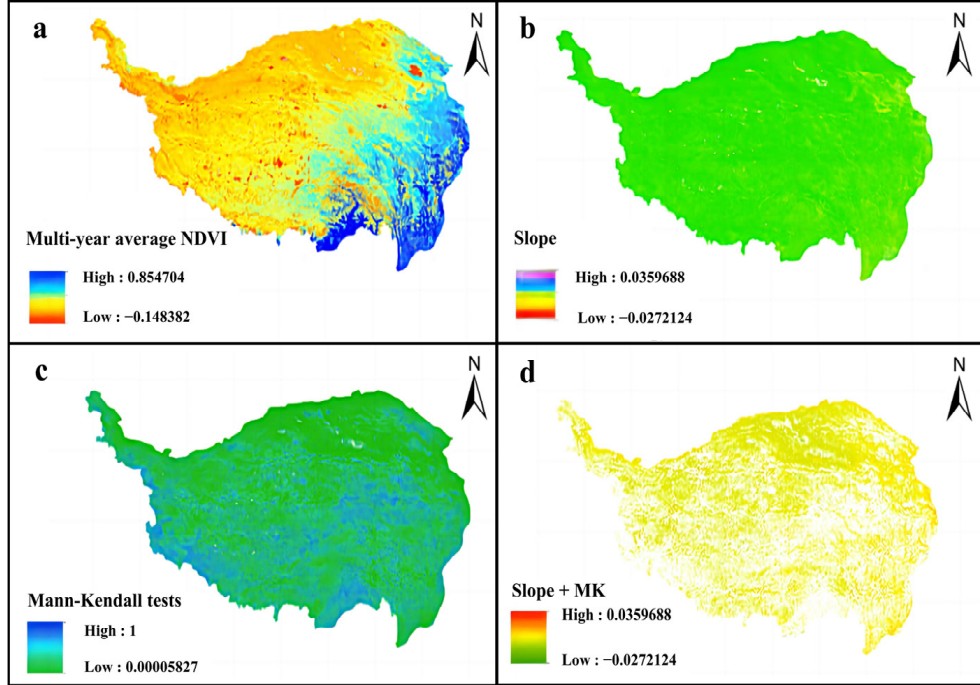

**Figure 7.** Spatiotemporal variation in NDVI over the Tibetan Plateau during 2000–2020. (**a**) Spatial distribution of multi-year average NDVI values over the Tibetan Plateau during 2000–2020. (**b**) Sen's slope estimation of NDVI values over the Tibetan Plateau during 2000–2020. (**c**) Mann–Kendall tests of the NDVI over the Tibetan Plateau during 2000–2020. (**d**) The significance of NDVI changes over the Tibetan Plateau during 2000–2020.

### 3.2.2. Effects of Climate Change on NDVI Trends in the Tibetan Plateau from 2000 to 2020

From 2000 to 2020, there was a significant positive correlation between the air temperature and the multi-year average NDVI values in the central, southern, and eastern regions of the Tibetan Plateau, while a small portion of the southwestern and central regions were negatively correlated with the air temperature. The other regions showed no obvious relationship between the multi-year average NDVI values and air temperature. In the eastern, southwestern, and northwestern regions of the Qinghai–Tibet Plateau, there is a significant positive correlation between the average NDVI value and precipitation. Conversely, in the southern and central areas of the region, the average NDVI value has a negative correlation with precipitation (Figure 8).

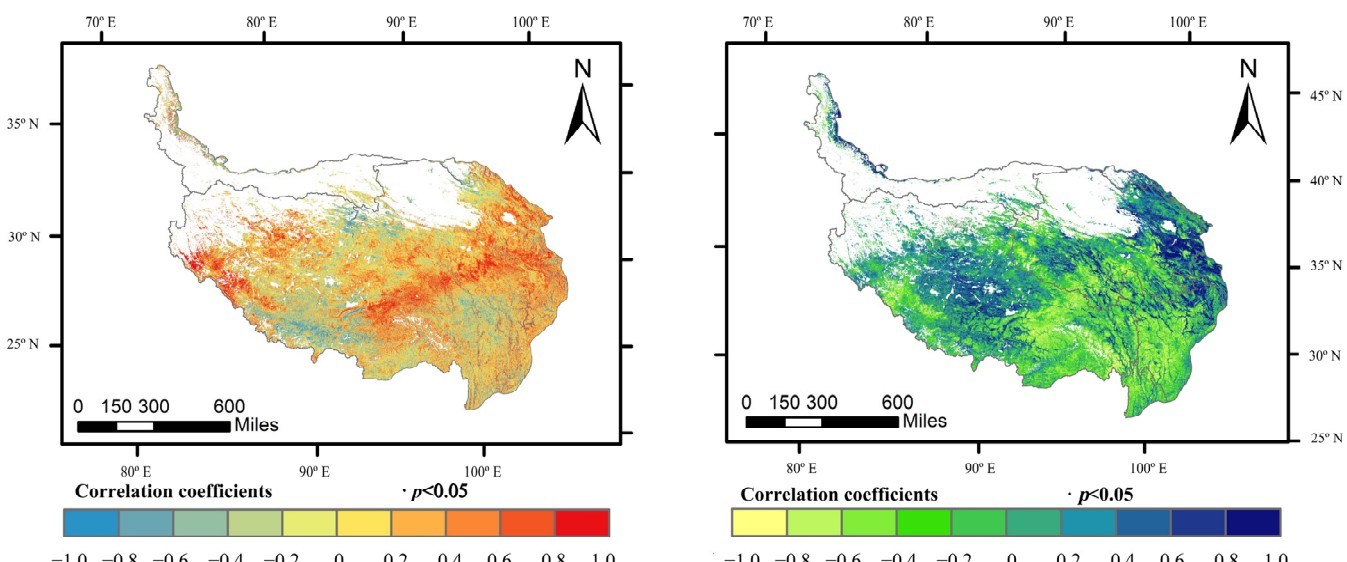

**Figure 8.** Spatial distribution of correlation between average NDVI values, and temperature and precipitation, over the Tibetan Plateau during 2000–2020.

### 3.2.3. Trend of LAI Change in the Tibetan Plateau from 2000 to 2020

During the period from 2000 to 2020, overall, the LAI values on the Tibetan Plateau showed an increasing trend, which was in line with the trend of NDVI values during the study period. Additionally, the LAI values did not show a significant increase before 2015; however, they showed a significant increase after 2015 (Figure 9). The LAI values on the Tibetan Plateau from 2000 to 2020 ranged from zero to seven, and the spatial heterogeneity of the multi-year mean LAI values was clear, with most regions of the Tibetan Plateau having low LAI values and only a small number of high-value areas distributed in the southern region. The LAI values in the eastern region were higher compared to those in the western region, with an overall gradual decrease from the southeast to the northwest (Figure 10a). The LAI values exhibited a general upward trend, with the trend being more pronounced in Northeast China. The MK test found that the overall changes in the northeastern region of the Tibetan Plateau were more significant compared to those of the southern region (Figure 10b–d).

### 3.2.4. Effects of Climate Change on LAI Trends of the Tibetan Plateau from 2000 to 2020

From 2000 to 2020, the multi-year average LAI values in the central part and a small area of the southwestern part of the Tibetan Plateau were significantly positively correlated with temperature. Most of the eastern part was positively correlated, and most of the southwestern part was negatively correlated with temperature. Some parts of the northwestern and northeastern parts showed a significant positive correlation with precipitation, and most of the Tibetan Plateau showed a negative correlation with precipitation (Figure 11).

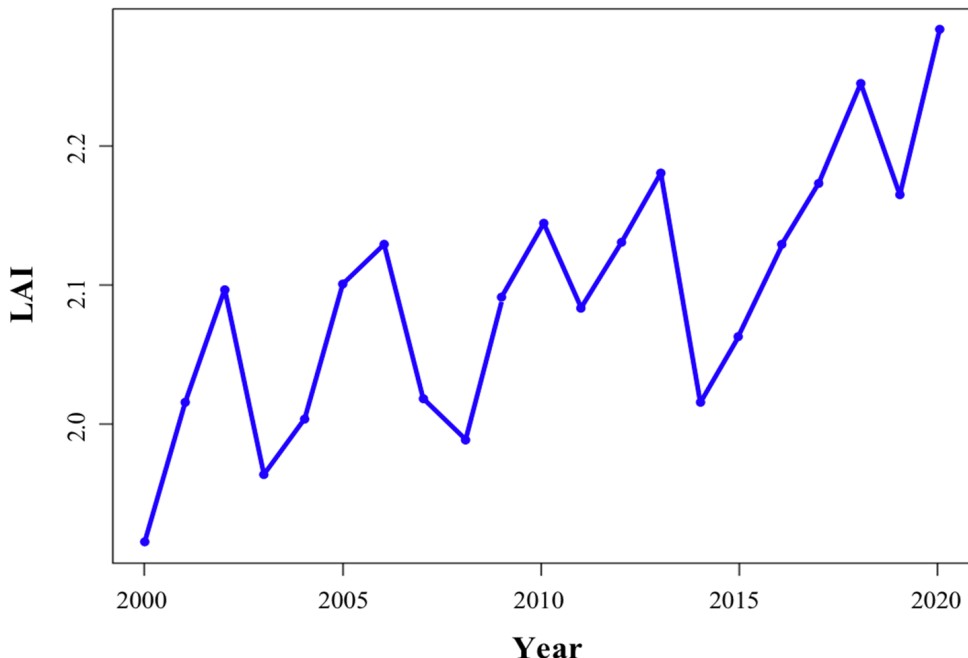

**Figure 9.** Changes in LAI values over the Tibetan Plateau during 2000–2020.

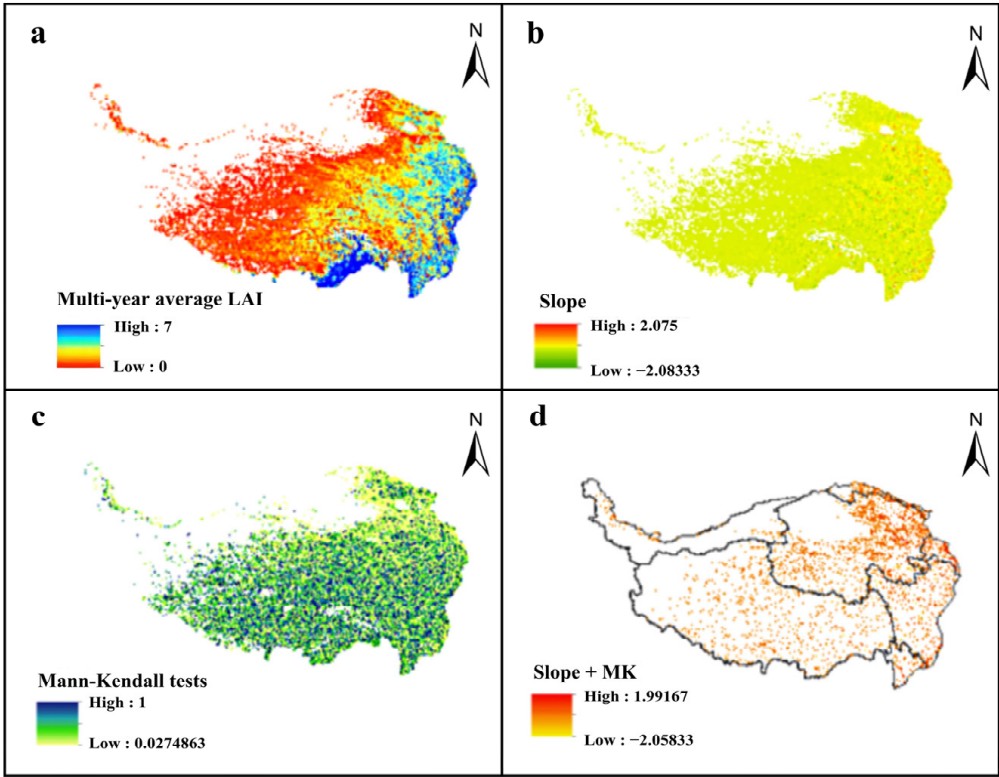

**Figure 10.** Spatiotemporal variation in LAI values over the Tibetan Plateau during 2000–2020. (**a**) Spatial distribution of multi-year average LAI values over the Tibetan Plateau during 2000–2020. (**b**) Sen's slope estimation of LAI values over the Tibetan Plateau during 2000–2020. (**c**) Mann–Kendall tests of LAI values over the Tibetan Plateau during 2000–2020. (**d**) The significance of LAI changes over the Tibetan Plateau during 2000–2020.

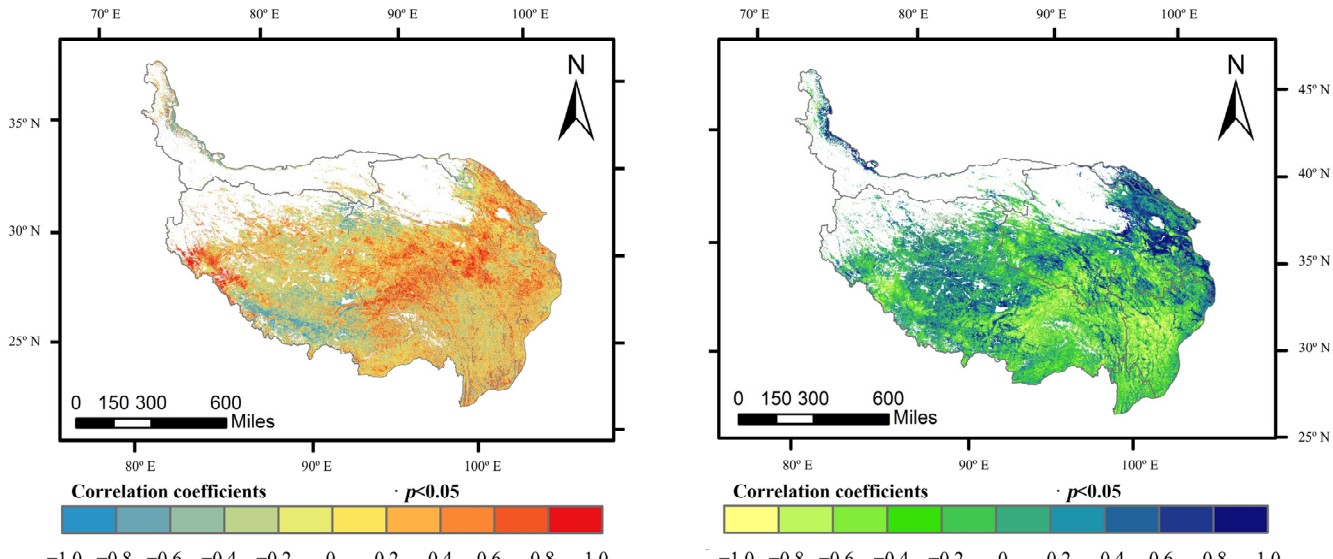

**Figure 11.** Spatial distribution of correlation between average LAI values, and temperature and precipitation, over the Tibetan Plateau during 2000–2020.

### 3.3. Climate Change Trends of the Tibetan Plateau from 2021 to 2100

The average predicted temperature of the Tibetan Plateau in the future period between 2021 and 2100 exhibited an upward trend year by year, and the increase was more obvious, with small fluctuations during the period (Figure 12a). From 2021 to 2040, the temperature in most areas of the central Tibetan Plateau showed an ascending trend, and the fluctuations in temperature were more obvious. Compared with the previous period, the temperature in most areas of the Tibetan Plateau region was decreasing, the temperature in some areas of the west was increasing, and the temperature fluctuation area changed to the northwest. Temperature fluctuations in the central and southeastern Tibetan Plateau were more obvious for the period between 2061 and 2080, and the temperature in most of the southwestern Tibetan Plateau was on an ascending trend. In contrast, from 2081 to 2100, temperature in most areas of the Tibetan Plateau exhibited a downward trend, while the western part experienced an increasing trend, with significant changes observed in the central part of the plateau, as well as fluctuations in the northern and western parts of the plateau (Figure 13).

The average predicted precipitation of the Qinghai–Tibet Plateau for the future period from 2021 to 2100 also showed an overall ascending trend year by year; however, it was smaller and more stable than the increase in temperature, with small fluctuations during the period (Figure 12b). The vast majority of the Qinghai–Tibet Plateau showed an ascending trend in precipitation for the period between 2021 and 2040. In the two periods from 2041 to 2060 and from 2081 to 2100, most of the Tibetan Plateau showed a downward trend in precipitation. Fluctuations in precipitation were more pronounced in the central part of the Qinghai–Tibet Plateau from 2041 to 2060, and in the eastern part of the Qinghai–Tibet Plateau from 2061 to 2080. The significance of precipitation changes in the same region of the Tibetan Plateau from 2021 to 2100 varied from one period to the next (Figure 14).

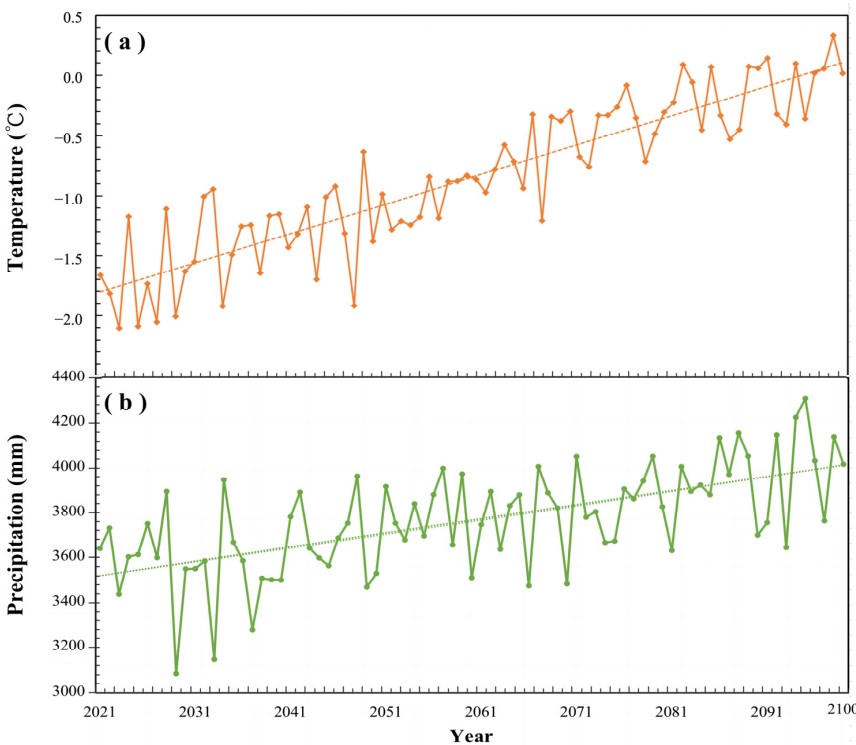

**Figure 12.** Interannual variations of mean temperature (**a**) and mean precipitation (**b**) over the Tibetan Plateau from 2021 to 2100.

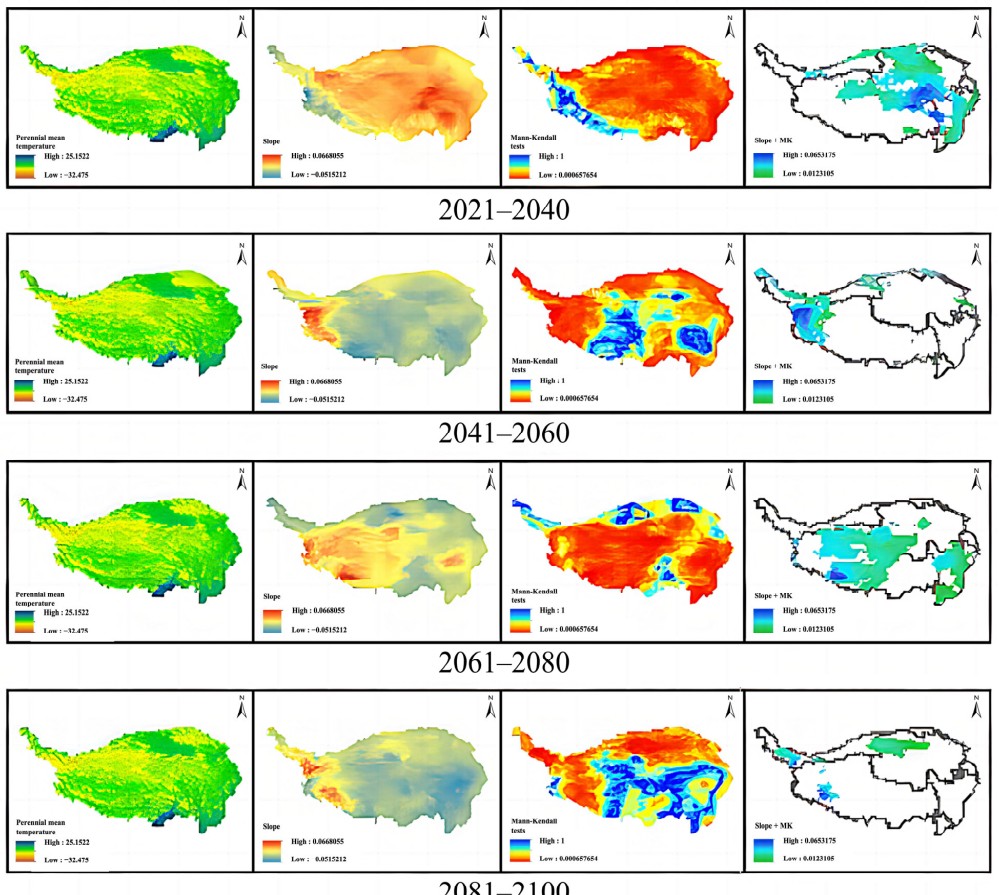

**Figure 13.** Temperature change trend of the Tibetan Plateau in the next four periods.

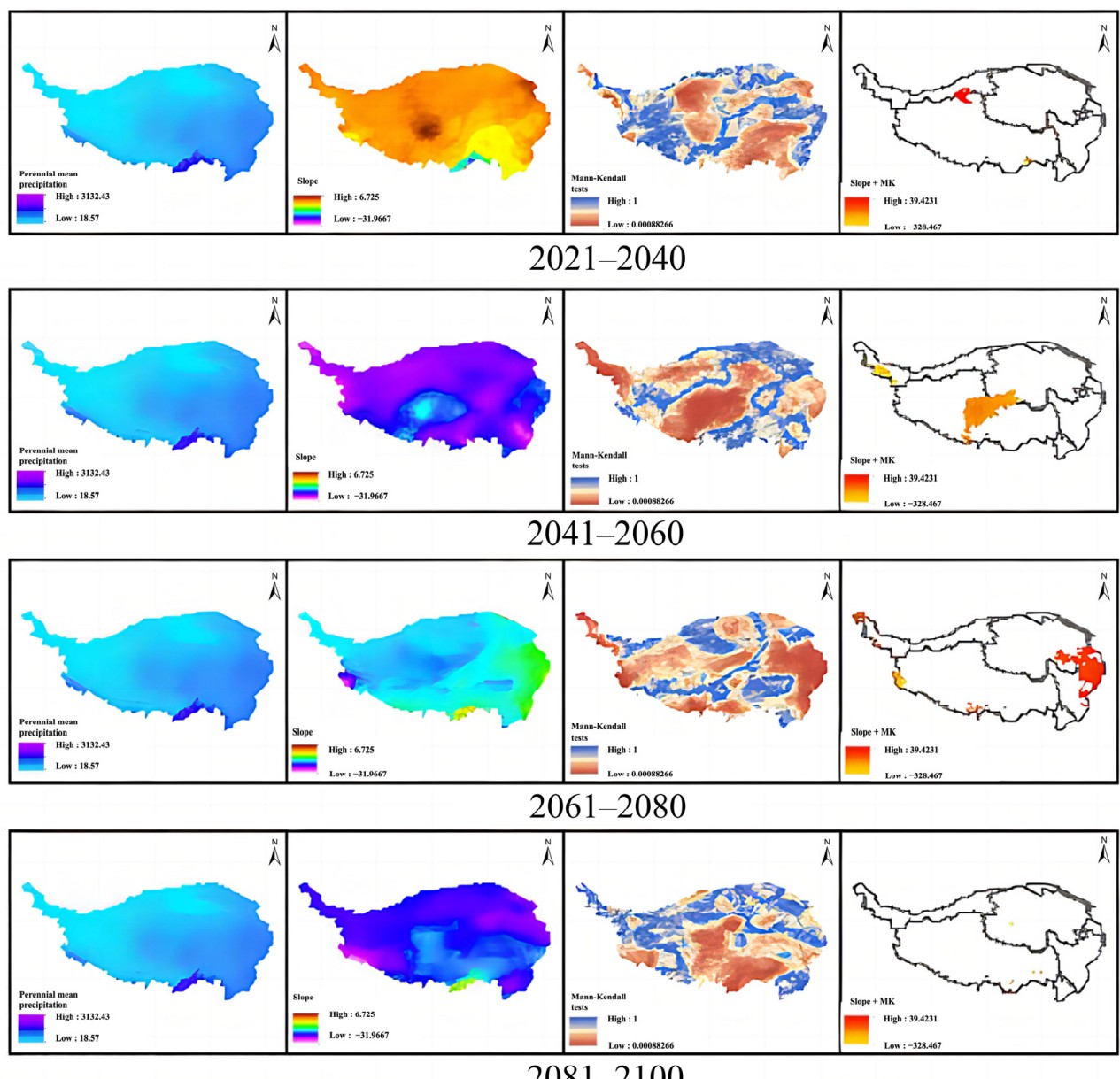

**Figure 14.** The precipitation change trend of the Tibetan Plateau in the next four periods.

### 3.4. Trends of Ecosystem Changes of the Tibetan Plateau from 2021 to 2100

The predicted average NDVI value of the Qinghai–Tibet Plateau for the future period from 2021 to 2100 showed an overall trend of ascending year by year, which was more obvious, with small fluctuations during the period, among which changes in the NDVI values were stable in the years from 2035 to 2045 and from 2050 to 2065 (Figure 15a). The average LAI value of the Tibetan Plateau for the future period between 2021 and 2100 showed an upward trend year by year, which was more obvious, with small fluctuations during the period, among which changes in the LAI values in the years from 2035 to 2045 and from 2050 to 2065 were more stable (Figure 15b). The trend change characteristics of the NDVI and LAI values were similar to each other, and also correlated with the trend changes in air temperature and precipitation to a certain extent. The NDVI and LAI values were more influenced by air temperature, but less affected by precipitation.

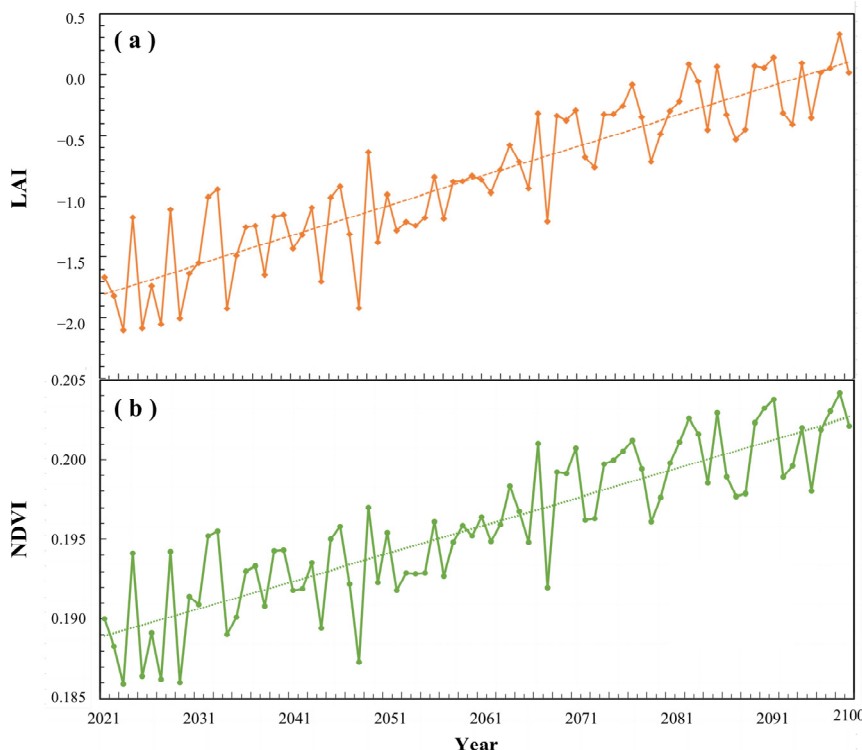

**Figure 15.** Interannual variations of mean LAI (**a**) and mean NDVI (**b**) values over the Tibetan Plateau from 2021 to 2100.

## 4. Discussion

This study highlights a significant increase in temperature and precipitation on the Tibetan Plateau between 2000 and 2020, indicating a clear pattern of warming and heightened humidity. These findings align with previous research that also documented a consistent rise in temperature and precipitation, demonstrating a substantial warming and moistening trend in the overall climate of the Tibetan Plateau in recent years [39–41]. There was some spatial heterogeneity in the precipitation and temperature changes on the Tibetan Plateau [42]. The overall temperature showed a gradual decrease from the southeast to the northwest, which was consistent with the results of [43]. The warming trends in the central and northeastern regions were significantly higher than those in other regions. Twenty years ago, precipitation increased in most areas of the Tibetan Plateau, and there was a significant increase in precipitation in the eastern region and a significant decrease in precipitation in some parts of the southern region. Consistent with the spatial distribution of temperature, the distribution of precipitation on the Tibetan Plateau showed a declining trend from the southeast to the northwest. The southern region had the most abundant precipitation, which was influenced by the topography of the plateau and monsoons [44]. Therefore, overall, the climate of the Tibetan Plateau showed warm and humid distribution characteristics in the southeast, and dry and cold characteristics in the northwest.

Temperature and precipitation were identified as the key drivers of spatial and temporal changes in vegetation in regions with a fragile ecological environment [45,46], and there are significant differences in the spatiotemporal distribution of the responses of different vegetation types to climate factors on the Tibetan Plateau [47]. Based on the results of this study, the overall trend of the NDVI and LAI values on the Tibetan Plateau demonstrated an ascending trend from 2000 to 2020 [12,43], which was the same as the trend for the temperature and precipitation changes. The spatial distribution of NDVI values on the Tibetan Plateau also showed a decreasing trend from the southeast to the northwest [48]. The significant increase in LAI values in the northeast of the Tibetan Plateau is consistent with the significant rise in air temperature in this region, which extends the duration of the vegetation growing season [49], improves productivity [50], and promotes the growth of

plateau vegetation [51]. The average multi-year NDVI values in most parts were influenced by both precipitation and air temperature. In particular, the average multi-year NDVI values in certain areas of the southern and eastern regions exhibit a significant positive correlation with air temperature. Additionally, the average multi-year LAI values in most of the eastern regions show a positive correlation with air temperature, which aligns with previous research findings [52]. However, there are some southwest and central regions where the NDVI values are negatively correlated with air temperature. This could be attributed to higher regional altitudes, leading to faster warming [53], increased vegetation respiration intensity [54], and the evaporation of soil surface water [55], thus limiting vegetation growth. In terms of precipitation effects, both the NDVI and LAI values exhibited negative correlations in certain southern and central regions. This can be explained by the fact that increased precipitation reduces both plant photosynthesis intensity and temperatures [56], thereby limiting plant growth to a certain extent. On the other hand, specific regions in the northwest and northeast showed a significant positive correlation between precipitation and LAI values [57]. This might be attributed to the relatively dry climates within these areas, where increased precipitation provides necessary water for plant growth, resulting in notable changes in plant development within this region [58,59]. The trend changes in the LAI and NDVI values in different regions of the Tibetan Plateau due to climate warming and humidification explained the different response mechanisms of different vegetation types to climate change, and further reflected the influence of climate factors on the ecosystem of the Tibetan Plateau.

Based on the correlation between precipitation, temperature, NDVI values, LAI values, and their effects, a prediction of the climate and ecological environment of the Tibetan Plateau in the next 80 years (2021–2100) was performed using multiple linear regression modeling. A significant warming and humidifying trend was found for the climate of the Qinghai–Tibet Plateau, with small fluctuations during the period, as well as the fact that the magnitude of changes due to increases in precipitation would be smaller than that of increases in temperature. This result was consistent with the prediction of GAO et al. [60]. The trend of temperature change at the same site varied at different periods, with different degrees of significance; however, the general pattern of climate change was the same and synchronized, which confirms previous climate predictions of the Tibetan Plateau [61]. The LAI and NDVI values showed a rising trend, which is the same as that of Ouyang et al., who predicted that the LAI values of the Qinghai–Tibet Plateau would increase in the future, and found that the vegetation quality would increase [62]. The NDVI and LAI are similar to each other and correlate with temperature and precipitation trends. Compared with the effect of precipitation on NDVI and LAI values, the effect of temperature on both values was higher. In addition, because temperature and precipitation are cumulative processes, there was a cumulative effect with the NDVI and LAI values, and the delay in vegetation response to climate factors may lead to experimental uncertainties. Huang et al. analyzed 17-year NDVI trends on the Tibetan Plateau and found that the lags in NDVI values on precipitation were more obvious, and the lags for air temperature were not obvious [19]. Among the climate drivers, in addition to precipitation and temperature as considered in this study, which may affect vegetation, other climate factors may also have an impact, such as solar radiation [63] and humidity [64]. Additionally, non-climatic factors, such as fewer meteorological observation stations [65], extreme disasters [14], and human activities [61] in the southwestern part of the Qinghai–Tibet Plateau may also have an impact on the results of this study. Therefore, in the future, more research is required to explore the impacts of both climatic and non-climatic factors on the ecological environment of the Qinghai–Tibet Plateau. This will help to enhance the monitoring and early warning capabilities for related climatic hazards and protect the fragile ecological environment of the Qinghai–Tibet Plateau region.

By studying and analyzing the impact of the vegetation system on the Tibetan Plateau in response to climate warming and humidification trends, as well as its future change projections, we can elucidate the response mechanisms of the Tibetan Plateau's vegetation

system to climate events. This research can provide a scientific foundation and reference for government departments to formulate and implement sustainable ecological management plans. These efforts aim to enhance vegetation's adaptability to climate change, reduce the inhibitory effects of climate disasters on its growth and development, and ultimately protect the ecological environment of the Tibetan Plateau.

## 5. Conclusions

In this study, we analyzed the climate change patterns of the Tibetan Plateau from 2000 to 2020 by collecting and organizing climate (temperature and precipitation) and ecosystem (NDVI and LAI) data, and by adopting a significance test based on a combination of the Mann–Kendall test and Sen's slope estimation. We revealed the effects of precipitation and temperature factors on the vegetation index of the ecosystems on the Tibetan Plateau, and a multiple regression model was constructed to predict the trends of LAI and NDVI values on the Qinghai–Tibet Plateau under climate change in the future from 2021 to 2100. The results showed the following:

(1) Spatial heterogeneity was observed in the precipitation and temperature changes across the Tibetan Plateau. Over the period from 2000 to 2020, both temperature and precipitation exhibited a general increase, indicating a clear tendency towards climate warming and increased humidity. Two decades ago, precipitation levels rose across most regions of the Tibetan Plateau, with a significant increase in the eastern part and a notable decrease in certain areas of the plateau.

(2) Different regions of the Tibetan Plateau exhibited varying correlations between ecosystems and the climate. From 2000 to 2020, there was a consistent upward trend in NDVI and LAI values across the Tibetan Plateau, in line with the trends of temperature and precipitation changes. In addition, the spatial distribution of NDVI values on the Qinghai–Tibet Plateau indicated a declining trend from the southeast to the northwest. The multi-year average NDVI values in the southern and eastern regions displayed a significant positive correlation with air temperature, while the LAI values in the northwest and northeastern areas showed a notable positive correlation with precipitation.

(3) The future climate of the Qinghai–Tibet Plateau showed a significant warming and humidification trend. While there were minor fluctuations during this period, the changes in precipitation were less pronounced than the changes in temperature. The temperature trends varied at the same location in different time periods, with varying degrees of significance. However, the overall climate change trend remained consistent, with synchronous changes observed. Both the NDVI and LAI values displayed an upward trend, with a stronger similarity observed between the NDVI and LAI values. Additionally, there was a correlation with the trends of the temperature and precipitation changes. Notably, the impact of air temperature on both NDVI and LAI values was found to be greater when compared to the influence of precipitation.

**Author Contributions:** Conceptualization, Z.H., T.Z., T.Y., J.W. and W.Y.; methodology, Z.H. and T.Y.; validation, T.Y. and Y.F.; formal analysis, Z.H., Y.P. and T.Y.; investigation, Z.H., T.Z. and J.C.; resources, T.H.F. and X.W.; writing—original draft preparation, Z.H.; writing—review and editing, Z.H., T.Z., J.C., Y.F., Y.P., L.Z., T.H.F., X.W., J.W. and W.Y.; visualization, Z.H. and Y.F.; supervision, T.Z., T.H.F., X.W., J.W. and W.Y.; project administration, Z.H., T.Z., T.H.F., X.W., J.W. and W.Y.; funding acquisition, J.W. and W.Y. All authors have read and agreed to the published version of the manuscript.

**Funding:** This study was supported by joint funds from the National Natural Science Foundation of China (U21A20187), the National Natural Science Foundation of China (42007383), the Hunan Water Conservancy Science and Technology Project (XSKJ2022068-35), and the follow up work of the Three Gorges Project of MWR (HY110161A0012022).

**Data Availability Statement:** The data presented in this study are available upon request from the corresponding author. The data are not publicly available due to the funded projects not yet being completed.

**Acknowledgments:** We would like to thank the National Tibetan Plateau Science Data Center for providing the China Monthly Precipitation Dataset and the China Monthly Mean Temperature Dataset (http://data.tpdc.ac.cn (accessed on 9 October 2022)). The MODIS NDVI dataset and MODIS LAI dataset were provided by the National Aeronautics and Space Administration (NASA) (https://ladsweb.modaps.eosdis.nasa.gov/search/ (accessed on 22 September 2022)). The MODIS NDVI and MODIS LAI datasets were provided by NASA.

**Conflicts of Interest:** The authors declare no conflict of interest.

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
