# Peer review of "Impacts of Climate Warming and Humidification on Vegetation Activity over the Tibetan Plateau"

_forests, doi:10.3390/f14102055_

Round 1

Reviewer 1 Report

Dear authors, 

Congratulations on your work. I have some suggestions to incorporate, and I have attached the file with more details. 

I suggest a change of title to " climate warming and humidification cause increased vegetation indexed in the Tibetan Plateau" (or something similar). 

The first paragraph of the introduction section is too long. I suggest shortening it. I have made some suggestions for eliminations. 

There are a lot of maps, I suggest representing only the most relevant information and reducing the number of figures. 

Also, you should eliminate repetitions. 

For the abbreviations (acronyms) you should spell them out, the first time you mention each one of them. 

The methods should be explained better as a lot of technicalities are used. 

In the analysis section, I don't think that all the formulae are needed, as they are difficult to understand. 

Use the same style for figures 2, 5, and 8. 

For figures that show tendency over time (3,6,9,13) I suggest only one figure that indicates increase, decrease, and stability between the years analyzed. 

kind regards

some minor revisiones are needed 

Reviewer 2 Report

Dear authors,

I would like to congratulate the authors. I reviwed the ms  "Impacts of climate warming and humidification on vegetation 2 activity (NDVI and LAI) over the Tibetan Plateau". The ms has an important data but it´s not ready. I have major suggestions:

L47-50: don´t you have recent data? (after 2012?);

Figure 1: provide full title, please (localization etc)

Figure 2/5/16 and others: provide units in y-axis

In my opinion, the ms has a lot of figures - 19!! I suggest to the authors choose the most important and insert some figures as a supplemental materials.

Discussion: could the authors explore the alternatives and impacts of the results for society and environment?

Keywords: provide words different from the title.

Round 2

Reviewer 2 Report

Dear authors,

Thank you for considerations. The authors improved the ms and in my opinion is ready to publish.